# Self-Adaptive Motion Tracking against On-body Displacement of Flexible Sensors

**Chengxu Zuo**[1]     **Jiawei Fang**[1]     **Shihui Guo**[1*]     **Yipeng Qin**[2]

[1]School of Informatics, Xiamen University, China
[2]School of Computer Science & Informatics, Cardiff University, UK

## Abstract

Flexible sensors are promising for ubiquitous sensing of human status due to their flexibility and easy integration as wearable systems. However, on-body displacement of sensors is inevitable since the device cannot be firmly worn at a fixed position across different sessions. This displacement issue causes complicated patterns and significant challenges to subsequent machine learning algorithms. Our work proposes a novel self-adaptive motion tracking network to address this challenge. Our network consists of three novel components: i) a light-weight learnable Affine Transformation layer whose parameters can be tuned to efficiently adapt to unknown displacements; ii) a Fourier-encoded LSTM network for better pattern identification; iii) a novel sequence discrepancy loss equipped with auxiliary regressors for unsupervised tuning of Affine Transformation parameters. Experimental results show that our method is robust across different on-body position configurations. Our dataset and code are available at: https://github.com/ZuoCX1996/Self-Adaptive-Motion-Tracking-against-On-body-Displacement-of-Flexible-Sensors.

## 1 Introduction

Following the philosophy of ubiquitous computing, wearable devices have emerged as a promising solution for motion tracking that has various applications in entertainment [1], human-computer interaction [2], healthcare [3, 4], etc. Among different types of such wearable devices, those using *flexible sensors* stand out for their advantages in long-term use scenarios. Specifically, flexible sensors can be easily integrated into wearable devices (*e.g.*, sewn into ordinary clothing) while ensuring wearing comfort. These flexible sensors bend in response to human body movements (represented by joint angles), causing changes in their readings, which can be used for motion tracking [5].

In recent years, enabling wearable devices to be worn in non-fixed positions has become a crucial objective in order to make these devices more suitable for daily use [6]. However, this flexibility poses new challenges for deep-learning powered motion capture (mocap) systems. That is, the inevitable on-body displacement across wearing sessions can cause significant data distribution shifts, which degrades the performance of supervised learning models (Figure 1a).

A straightforward solution to the above problem is to collect a large dataset that covers a wide spectrum of wearing positions for model training. However, this approach significantly amplifies data collection costs, including equipment expenses, manpower, and time investments. Furthermore, any alterations in device design (sensor position, quantity, etc.) necessitate a fresh round of data collection, resulting in significant resource consumption and impeding swift product iteration. Another solution is to fine-tune the model against "unseen" displacements with a small amount of labeled data under new displacements, *a.k.a.*, supervised domain adaptation. However, this is infeasible for our task as

---

*Corresponding author: Shihui Guo (guoshihui@xmu.edu.cn)

37th Conference on Neural Information Processing Systems (NeurIPS 2023).

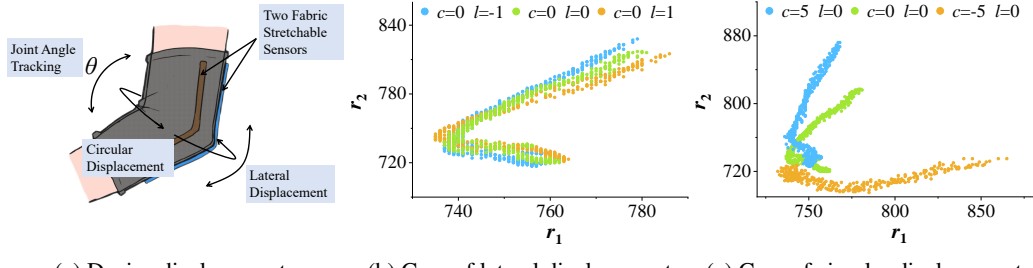

(a) Device displacements     (b) Case of lateral displacement     (c) Case of circular displacement

Figure 1: Demonstration of our problem setup. (a) Joint tracking with lateral and circular sensor displacements. (b) The data distribution shifts caused by lateral displacement, *e.g.,* $l$=1: lateral displacement shifts upwards by 1 cm; and (c) circular displacements, *e.g.,* $c$=5: circular displacement shifts clockwise by $5°$. $r_1$, $r_2$: sensor readings.

obtaining accurate joint angles (*i.e.*, labels) requires advanced optical motion capture systems that are difficult to set up in real-world scenarios (*e.g.,* outdoor running, boating), hindering everyday use.

In this work, we address the above challenges by proposing a novel self-adaptive motion tracking network that can adapt itself to unknown displacements in an unsupervised way, using only a small amount of new sensor readings. Specifically, most existing unsupervised domain adaptation methods assume the source and target domains share identical feature distributions, which is not applicable to our task as our training set contains data with multiple displacements, while the test data has only one unknown displacement. To this end, we propose a novel solution that allows the alignment of test data with a subset of training data that also has one displacement, which encompasses three novel components: i) a light-weight learnable Affine Transformation layer whose parameters effectively model the distributional shifts caused by displacements; ii) a Fourier-encoded LSTM network that facilitates the learning of more frequency components for better pattern learning and identification; iii) a novel sequence discrepancy loss equipped with auxiliary regressors for unsupervised self-adaptation.

The contributions of our method are three-fold:

- We propose a novel self-adaptive motion tracking network that can adapt to unknown on-body displacements of flexible sensors in an unsupervised way, allowing for long-term and daily motion tracking.

- We propose three novel components for our network: i) a light-weight learnable Affine Transformation layer, ii) a Fourier-encoded LSTM network, and iii) a novel sequence discrepancy loss, which together efficiently reduce the distributional shifts of sensor readings caused by on-body displacements.

- Extensive experiments demonstrate the superior performance of our method against the state-of-the-art methods of domain adaptation applied in our scenario.

## 2 Related Work

### 2.1 Domain Adaptation

Domain adaptation aims to mitigate the gap between the source and target domains so that models trained in the source domain(s) can be applied to the target domain(s). Traditional methods perform adaptation effectively by either reweighting samples from the source domain [7, 8], or seeking an explicit feature transformation that transforms the source and target samples into the same feature spaces [9, 10, 11, 12]. Subsequent studies have shown that deep neural networks can learn more transferable features for domain adaptation [13]. For example, Deep Domain Confusion (DDC) [14] first proposed the use of Maximum Mean Discrepancy (MMD) loss to align the feature distribution of the target domain with that of the source domain for the domain adaption of deep neural networks; Deep Adaptation Networks (DAN) [15] extends the idea to the use of multiple-kernel MMD; Deep CORAL [16] proposed CORAL loss [17] for adaptation; Chen *et al.*[18] propose a Higher-order Moment Matching (HoMM) method for better distribution discrepancy minimizing. Other works within deep neural networks also illustrate immense success in learning transferable

features [19, 20, 21]. In addition, recent studies show that adversarial learning also contributes to learning more transferable and discriminative representations [22, 23, 24, 25, 26, 27, 28].

In summary, most existing domain adaptation methods assume that the source and target domains share the same feature distributions and thus aim to minimize relevant distributional differences [29]. While in our task, the training set contains data with multiple displacements while the test data has only one unknown displacement, thus making the above-mentioned assumption invalid.

Addressing this issue, we propose a novel self-adaptive motion tracking network that tunes the parameters of an Affine Transformation layer with a novel sequence discrepancy loss, which can align the test data with a subset of training data that also has one displacement, achieving robust and accurate domain adaptation for flexible sensors.

## 2.2 Human Motion Capture

Human motion capture records human body movements and has been widely applied in entertainment, healthcare, sports, etc. Existing mocap solutions can be roughly classified into two categories: vision-based and sensor-based.

Vision-based mocap solutions make use of the latest deep learning techniques and have achieved great success in specific scenarios [30, 31]. However, they rely on good visual conditions and are inherently weak against textureless clothes and environmental problems (*e.g.*, challenging lighting, occlusion). Most existing sensor-based mocap solutions rely on Inertial Measurement Units (IMUs) to record motion inertia / acceleration for the analysis of human posture [32, 33, 34]. Although robust against environmental conditions, the dense and tight placement of IMUs is intrusive and inconvenient, hampering performers from moving freely in their daily lives.

To this end, people turned to flexible sensor-based wearable devices indistinguishable from daily clothing. For example, Glauser *et al.* [35] designed a stretch-sensing soft glove and used it to interactively estimate hand poses with the aid of a deep neural network; Ma *et al.* [36] proposed flexible all-textile dual tactile-tension sensors for precise monitoring of athletic and form, illustrating their potential application in robust physical training analysis. Zhou *et al.* [37] uses a deep regressor to continuously predict the 3D upper body joints coordinates from 16-channel textile capacitive sensors.

## 2.3 Flexible Sensors

Given their advantages of bio-compatibility, high stretch-ability, lightweight, and ease of integration within clothing, flexible sensors have been used for long-term monitoring of human physical status, specifically motion capture [38], human-computer interfaces [39], soft robotics [40], etc. For human motion tracking, existing methods have explored the use of flexible sensors in tracking the motion of the upper body [41], fingers [35], lower limbs [42], elbow joints [43], and knee joints [44]. Along with such exploration, it has been noted that on-body displacement of sensors is inevitable as the device cannot be firmly worn at a fixed position across different sessions [45]. Additionally, due to the deformation characteristics of flexible sensors, it is fairly complicated to achieve robust motion tracking in the presence of placement deviation [46, 47].

In this work, we start with an exemplary scenario: an elbow pad with two flexible and stretchable sensors for elbow joint tracking, and address this gap by proposing a novel self-adaptive motion tracking network that can adapt to unknown on-body displacements in an unsupervised manner to achieve robust and accurate motion tracking.

# 3 Hardware

We design and develop a prototype by augmenting a standard elbow pad with two soft stretchable (*i.e.*, flexible) sensors, which are placed on the olecranon side of the elbow. The pad offers a versatile size of 20 cm in length and 25 cm in circumference, accommodating a broad spectrum of users. The two sensors are placed 2 cm apart.

Our method aims to estimate the bending angle, $\theta$, of an elbow joint (Figure 1) from the sensor readings of the two flexible sensors. The bending angle is defined as the angle in the sagittal plane

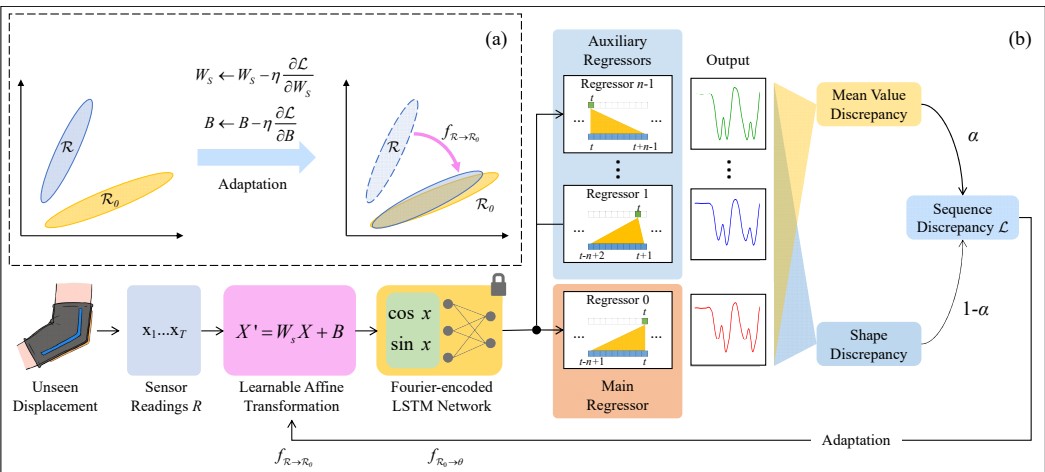

Figure 2: Overview of the proposed self-adaptive motion tracking network. Our network decomposes the mapping $f$ between sensor readings $\mathcal{R}$ and joint angles $\theta$ into two parts: $f_{\mathcal{R}\to\theta} = f_{\mathcal{R}\to\mathcal{R}_0} \cdot f_{\mathcal{R}_0\to\theta}$. (a) The intuition behind the adaptation function $f_{\mathcal{R}\to\mathcal{R}_0}$, which aligns the test data $\mathcal{R}$ to a subset of the training set $\mathcal{R}_0$ that also has a single displacement. (b) Illustration of the three novel components in the proposed network: i) a learnable Affine Transformation layer; ii) a Fourier-encoded LSTM network; iii) a sequence discrepancy loss $\mathcal{L}$ equipped with auxiliary regressors. Specifically, $f_{\mathcal{R}\to\mathcal{R}_0}$ is implemented with the Affine Transformation layer, whose parameters are adapted with the sequence discrepancy loss $\mathcal{L}$; $f_{\mathcal{R}_0\to\theta}$ is implemented with the supervisedly pre-trained Fourier-encoded LSTM network, whose parameters are frozen during adaptation.

between the humerus and the central line between the radius and the ulna. Fabric sensors are purchased as off-shelf products from ElasTech[1]. They are capacitive, *i.e.*, their capacitance increase with the stretch caused by the bending of the arm. The sensor readings are digitized to values in the range [0, 1023] and transmitted wirelessly via Bluetooth Low Energy at a frame rate of 50Hz.

## 4   Method

In our work, a motion tracking system using flexible sensors aims to construct a function $f$ between sensor readings $R$ and joint angles $\theta$ (Figure 1a) that: $\theta = f(R)$. Naively, $f$ can be approximated through supervised learning by fitting a model to collected $(R, \theta)$ data. While in practice, it is infeasible to collect enough $(R, \theta)$ data for training as $R$ depends on the countless instances of on-body displacement of flexible sensors $(c, l)$, where $c$ (deg) and $l$ (cm) are circular and lateral displacement, respectively (Figure 1a). Addressing this issue, we propose a novel self-adaptive motion tracking network for flexible sensors that can efficiently adapt to unseen on-body displacements in an unsupervised manner. As Figure 2 shows, our network decomposes $f$ into two parts:

$$f_{\mathcal{R}\to\theta} = f_{\mathcal{R}\to\mathcal{R}_0} \cdot f_{\mathcal{R}_0\to\theta} \tag{1}$$

where $\mathcal{R}_0$ denotes the sensor readings with a limited number of displacements in the training dataset, $\mathcal{R}$ denotes the sensor readings with unknown displacements at the time of testing. Between them, $f_{\mathcal{R}\to\mathcal{R}_0}$ is implemented with a light-weight learnable Affine Transformation layer whose parameters can be tuned to align $\mathcal{R}$ to $\mathcal{R}_0$ and thus adapt the network to "unseen" displacements (Sec. 4.1); $f_{\mathcal{R}_0\to\theta}$ is implemented with a Fourier-encoded LSTM network to reduce the learning bias towards low-frequency functions and produce more accurate results (Sec. 4.2). The tuning of $f_{\mathcal{R}\to\mathcal{R}_0}$ is achieved by minimizing a novel sequence discrepancy loss equipped with auxiliary regressors (Sec. 4.3).

---

[1] http://www.elas-tech.com/

## 4.1  Learnable Affine Transformation

The proposed learnable Affine Transformation layer is an efficient implementation of $f_{\mathcal{R} \to \mathcal{R}_0}$. Specifically, it models the *linear* transformation components of flexible sensors in a motion tracking system:

- **Initial Stretch**, which is mainly affected by lateral displacements (Figure 1(b)) and can be modeled with the additive **bias** vector in affine transformation.

- **Stretching Range**, which is mainly affected by circular displacement (Figure 1(c)) and can be modeled with a **scaling** matrix in affine transformation.

Accordingly, let $X = [r_1, r_2]^T$ be the sensor readings, $W_s = diag\,[s_1, s_2]$ be the scaling matrix, $B = [b_1, b_2]^T$ be the bias vector, we have:

$$f_{\mathcal{R} \to \mathcal{R}_0}(X) = W_s X + B = \begin{bmatrix} s_1 & 0 \\ 0 & s_2 \end{bmatrix} \begin{bmatrix} r_1 \\ r_2 \end{bmatrix} + \begin{bmatrix} b_1 \\ b_2 \end{bmatrix} \tag{2}$$

Both $W_s$ and $B$ are learnable in the adaptation stage (Sec. 4.3) for the adaptation to unknown displacements.

## 4.2  Fourier-encoded LSTM Network

We implement $f_{\mathcal{R}_0 \to \theta}$ by augmenting a standard LSTM sequence prediction network with a Fourier Feature Encoding (FFE) layer, which models the *nonlinear* mapping from flexible sensor signals to joint angles. Our key insight is that FFE helps to reduce the learning bias of deep neural networks towards low-frequency functions [48] that leads to underfitting of $f_{\mathcal{R}_0 \to \theta}$.

The FFE layer used in our work is as follows:

$$x' = \left[ \cos(\frac{2\pi x}{A}), \sin(\frac{2\pi x}{A}) \right] \tag{3}$$

where $x \in X$ is the input, $A > |\max(X) - \min(X)|$ so that $x'$ and $x$ are in one-to-one correspondence. Please note that our Fourier-encoded LSTM network is only trained in the pretraining stage and fixed during adaptation (Figure 2).

## 4.3  Unsupervised Adaptation to Displacements

---
**Algorithm 1** Unsupervised Adaptation to Displacements

---
**Input**: Pretrained model $f$ with auxillary regressors; Scaling matrix $W_s$ and Bias vector $B$ of the affine transformation layer in $f$; Test-time sensor readings $R$; Batch size $n$;
**Parameter**: Training epoch $e$; Learning rate $\eta$;
**Output**: Adapted $W_s$ and $B$;
1: **for** $i = 1, \ldots, e$ **do**
2:     **for** $j = 1, \ldots, \lceil |R|/n \rceil$ **do**
3:         $X \leftarrow$ a mini-batch of (sequence) samples from $R$
4:         $\hat{\Theta} \leftarrow f(X)$
5:         $W_s \leftarrow W_s - \eta \frac{\partial \mathcal{L}(\hat{\Theta})}{\partial W_s}$ ; $B \leftarrow B - \eta \frac{\partial \mathcal{L}(\hat{\Theta})}{\partial B}$
6:     **end for**
7: **end for**
8: **return** $W_s$, $B$

---

As mentioned above, our network decomposes the mapping $f$ into two parts $f_{\mathcal{R} \to \theta} = f_{\mathcal{R} \to \mathcal{R}_0} \cdot f_{\mathcal{R}_0 \to \theta}$. Between them, the challenging nonlinear components are well-approximated by $f_{\mathcal{R}_0 \to \theta}$, which greatly simplifies our adaptation to tuning the parameters of the Affine Transformation layer of $f_{\mathcal{R} \to \mathcal{R}_0}$. Intuitively, we aim to align the distribution of $\mathcal{R}$ to that of $\mathcal{R}_0$ so that $f_{\mathcal{R}_0 \to \theta}$ can accurately map $\mathcal{R}$ to joint angle $\theta$. Naively, this can be achieved by directly minimizing a distributional difference

loss between $\mathcal{R}_0$ and $\mathcal{R}$ or maximizing the prediction accuracy when using $\mathcal{R}$. However, neither approach is effective for our task as i) the former assumes $\mathcal{R}$ and $\mathcal{R}_0$ share the same distribution after the affine transformation, which is incorrect because $\mathcal{R}_0$ (the training set) contains data with multiple displacements while $\mathcal{R}$ is collected with a single unknown displacement; and ii) the latter requires ground truth joint angles $\theta$ of $\mathcal{R}$, which are unavailable during use. Addressing this challenge, we propose a novel sequence discrepancy loss equipped with auxiliary regressors as follows.

**Sequence Discrepancy Loss.** We observed that the domain gaps among different displacements are particularly evident when the elbow is bent and less when the elbow is flexed. Consequently, the domain gaps are not evenly distributed over time during elbow movement, leading to the discrepancy among different regressors that estimate the same joint angle $\theta_t$ with different choices of time windows, where $t$ denotes the time step in a motion sequence. Our key idea is that the adaptation to unknown displacements can be achieved by minimizing such discrepancy. Specifically, let $n$ be the length of the time window for the main regressor, we have

$$\hat{\theta}_t^{(0)} = f(R_{t-n+1}, R_{t-n+2}, ..., R_t) \tag{4}$$

where $\hat{\theta}_t^{(0)}$ is its joint angle estimate. Similarly, we include additional $n-1$ auxiliary regressors that predict $\theta_t$ with different time windows as:

$$\hat{\theta}_t^{(i)} = f(R_{(t+i)-n+1}, R_{(t+i)-n+2}, ..., R_{t+i}) \tag{5}$$

where $\hat{\theta}_t^{(0)} \approx \hat{\theta}_t^{(1)} \approx ... \approx \hat{\theta}_t^{(n-1)}$ for $\mathcal{R}_0$ used in supervised pretraining, $i=1,...,n\text{-}1$. However, when $\mathcal{R}$ is used, discrepancies occur among regressors and we tune the affine transformation parameters (Sec. 4.1) to minimize such discrepancies. Specifically, let $\overline{\Theta}^{(k)} = \text{mean}(\hat{\theta}_1^{(k)}, \hat{\theta}_2^{(k)}, \dots, \hat{\theta}_t^{(k)})$ be the mean of joint angle estimates of regressor $k$, $\hat{\phi}_t^{(k)} = \frac{\hat{\theta}_t^{(k)} - \min_t(\hat{\theta}^{(k)})}{\max_t(\hat{\theta}^{(k)}) - \min_t(\hat{\theta}^{(k)}) + \varepsilon}$ be the normalized joint angles, $\varepsilon = 1$ for numerical purpose, we define:

$$\mathcal{L}_{mean} = \sum_{k=0}^{n-1}(\overline{\Theta}^{(k)} - \frac{1}{n}\sum_{i=0}^{n-1}\overline{\Theta}^{(i)})^2 \tag{6}$$

$$\mathcal{L}_{shape} = \sum_{t}\sum_{k=0}^{n-1}(\hat{\phi}_t^{(k)} - \frac{1}{n}\sum_{i=0}^{n-1}\hat{\phi}_t^{(i)})^2 \tag{7}$$

Our overall sequence discrepancy loss is a weighted sum of $\mathcal{L}_{mean}$ and $\mathcal{L}_{shape}$:

$$\mathcal{L} = \alpha\mathcal{L}_{mean} + (1-\alpha)\mathcal{L}_{shape} \tag{8}$$

where $\alpha \in [0, 1]$ is a weighting hyperparameter. Alg. 1 shows the pseudo-code of our adaptation algorithm.

# 5 Experimental Results

## 5.1 Experimental Setup

**Dataset and Metrics** The dataset used in this paper consists of sensor readings and joint angles collected by a single user wearing the augmented elbow pad (Sec. 3) while performing elbow flexion. The joint angles are computed using the 3D positions captured by a Nokov motion-capture system at a rate of 60 FPS. As instances of different on-body displacements, we collect 11 groups of data with circular displacements $c \in \{-5, 0, 5\}$ and lateral displacements $l \in \{-2, -1, 0, 1, 2\}$. For each group, we collected data for 8 consecutive elbow flexion. In total, we collected 5,310 frames of data across the 11 valid groups. Among the 11 groups, we randomly selected 5 of them as the training set $\mathcal{D}_{train}$ and used the rest 6 groups as the test set $\mathcal{D}_{test}$. We have obtained *ethical approval* for the publication of both datasets and results.

We use the Mean Absolute Error (MAE) of predicted joint angles (degrees) as our main evaluation metric. For each experiment, we repeat 20 times and report their mean and standard deviation (std). Unless specified, we use the same setups for all experiments.

**Training Details** Our model is trained in two stages: supervised pretraining and unsupervised adaptation. For the supervised pretraining, we employ an MSE loss:

$$\mathcal{L}_{mse} = \frac{1}{n} \sum_{k=0}^{n-1} \sum_{t} (\hat{\theta}_t^{(k)} - \theta_t)^2 \tag{9}$$

where $\hat{\theta}_t^{(k)}$ is the output of regressor $k$ at time step $t$, $\theta_t$ is the corresponding ground truth joint angle. We use an Adam optimizer with a learning rate of $1e^{-3}$, $\beta_1 = 0.9$, $\beta_2 = 0.999$, and training epoch $e = 30$. For the adaptation, we use an Adam optimizer with a learning rate of $5e^{-3}$ and a weight decay of 0.001, $\beta_1 = 0.9$, $\beta_2 = 0.999$, and training epoch $e = 20$. We use $n = 10$ for both pretraining and adaptation. All experiments were conducted on a desktop PC with an AMD Ryzen 3950X CPU and an NVIDIA RTX 3080 GPU. Please see the supplementary materials for the details of network architectures.

## 5.2 Comparisons with SOTA

As Table 1 shows, we compare our method with state-of-the-art (SOTA) domain adaptation methods. For a fair comparison, we have adapted the official code provided by the authors to share the same input and output format as ours. Please see the supplementary materials for more details on the implementation of SOTA methods. It can be observed that our method outperforms all SOTA methods, which demonstrates its effectiveness.

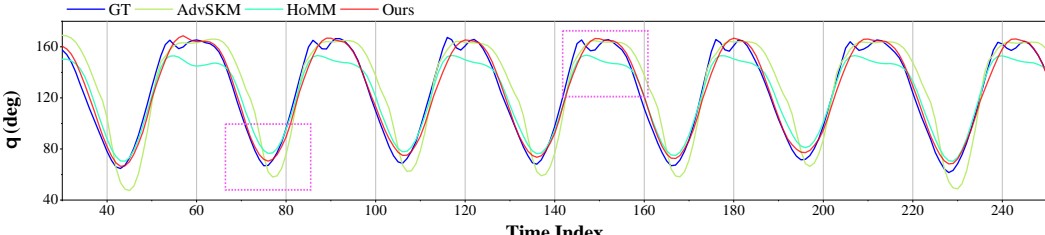

Figure 3: Comparison with two most competitive SOTA methods (displacement condition: $c$=-5, $l$=1). As highlighted in the dashed boxes, our method (red) is much closer to the ground truth (blue). GT: ground truth.

To gain more insights, we visualize the results of our method and its two strongest competitors, AdvSKM [27] and HoMM [18], as time series in Figure 3. The results show that our method fits the ground truth with a closer distance than other SOTA methods.

**Comparison with Supervised Learning** To demonstrate the effectiveness of unsupervised learning in our task, we tune the parameters of Affine Transformation layer by supervised learning (*i.e.*, supervised domain adaptation) to obtain its "upper bound". Even trained in an unsupervised manner, our method achieves results comparable to its supervised version (Table 1). This difference of 1.5 degrees indicates that our unsupervised learning approach is effective.

Table 1: Results of comparative experiments.

| Method | MAE (deg) |
|---|---|
| MMD [49] | $28.73 \pm 8.50$ |
| D-CORAL [16] | $13.83 \pm 8.45$ |
| DAAN [23] | $16.80 \pm 7.88$ |
| DSAN [50] | $14.03 \pm 5.88$ |
| AdvSKM [27] | $13.05 \pm 5.19$ |
| HoMM [18] | $11.03 \pm 4.42$ |
| Ours | $\mathbf{7.32 \pm 2.85}$ |
| Supervised | $5.63 \pm 1.95$ |

**Experiments on different participants** To further demonstrate the generalization ability of our method, we conducted experiments with five additional participants of varying body types (see the supplementary materials for body profile details). Each of these five participants wore the devices in three supervised on-body positions and performed three distinct physical activities sequentially: **ping-pong**, **basketball**, and **boxing**. Each activity lasted for a minimum duration of one minute. In total, this comprehensive evaluation approach yielded 15 (5 participants × 3 on-body displacements) unique data segments across all participants, comprising 81,848 total frames. For each participant,

Table 2: Experimental results on five additional participants.

| User ID | Averaged MAE (deg) | | | | |
| --- | --- | --- | --- | --- | --- |
| | D-CORAL | DSAN | HoMM | AdvSKM | Ours |
| 1 | **11.84 ± 1.49** | 11.86 ± 1.57 | 11.96 ± 1.73 | 11.84 ± 1.80 | 12.07 ± 1.04 |
| 2 | 16.59 ± 1.81 | 17.04 ± 1.80 | **16.53 ± 1.70** | 16.63 ± 0.95 | 16.69 ± 2.34 |
| 3 | 14.94 ± 5.89 | 13.68 ± 4.19 | 15.24 ± 6.93 | 15.71 ± 6.89 | **12.70 ± 2.79** |
| 4 | 11.76 ± 1.11 | 12.04 ± 1.16 | 12.06 ± 1.49 | 12.08 ± 1.73 | **9.30 ± 2.77** |
| 5 | 19.41 ± 4.37 | 19.32 ± 4.34 | 20.58 ± 5.88 | 19.44 ± 4.21 | **13.96 ± 2.88** |
| Avg | 14.91 ± 0.34 | 14.79 ± 0.22 | 15.27 ± 0.48 | 15.14 ± 0.18 | **12.94 ± 0.25** |

we conducted 5 times 3-fold cross-validation experiments and recorded the average result. As Table 2 shows, our method achieves optimal or near-optimal results for all five additional users, indicating the effectiveness of our method against user diversity.

## 5.3 Ablation Study

To demonstrate the effectiveness of the three novel techniques proposed, we conduct an ablation study as follows:

- **FFE**: Remove the Fourier Feature Encoding (FFE) layer from the LSTM network.
- **Affine**: Remove the Affine Transformation layer and tune all the parameters of the network during adaptation after supervised pretraining. For the adaptation, we use a weighted sum of two loss terms: i) a supervised loss $\mathcal{L}_{mse}$ applied on the training data; and ii) our sequence discrepancy loss $\mathcal{L}$ applied on the test data.
- **SD Loss**: Replace our Sequence Discrepancy (SD) loss with a more naive and stricter version:

$$\mathcal{L}^{'} = \sum_t \sum_{k=0}^{n-1} (\hat{\theta}_t^{(k)} - \overline{\Theta}_t)^2 \tag{10}$$

where $\overline{\Theta}_t = \mathrm{mean}(\hat{\theta}_t^{(0)}, ..., \hat{\theta}_t^{(n-1)})$.

Table 3: Ablation study.

| Case | FFE | Affine | SD Loss | MAE |
| --- | --- | --- | --- | --- |
| 1 | - | - | - | 13.40±5.58 |
| 2 | + | - | - | 11.81±6.65 |
| 3 | - | + | - | 12.58±8.62 |
| 4 | - | - | + | 14.36±5.96 |
| 5 | - | + | + | 10.27±4.20 |
| 6 | + | - | + | 12.36±6.99 |
| 7 | + | + | - | 9.38±8.78 |
| Ours | + | + | + | **7.32±2.85** |

We have enumerated all combinations of the three components proposed. As Table 3 shows, it can be observed that: i) Ours achieves the best performance, indicating the effectiveness of the three novel techniques proposed; ii) FFE and affine transformation improve the performance in all cases; iii) the effectiveness of SD loss relies on affine transformation. In both cases with and without FFE, the co-existence of affine transformation and SD loss show better performance than using SD loss alone (Ours over Case 6, Case 5 over Case 3).

## 5.4 Justification of Motivation

**Affine Transformation**    Figure 4 demonstrates the effective alignment of the target and source domains with our method. As shown in Figure 4a and 4b , although light-weight, the proposed Affine Transformation layer effectively aligns the sensor reading distributions of test data (blue) to those in the training set (orange), thereby significantly improving the accuracy of joint angle estimation.

**Sequence Discrepancy Loss**    The design rationale of this component is rooted in the insight that our sequence discrepancy loss $\mathcal{L}$ shares a similar optimization landscape with the estimation error of joint angles during optimization (Figure 5). This consistency effectively offers the advantage of

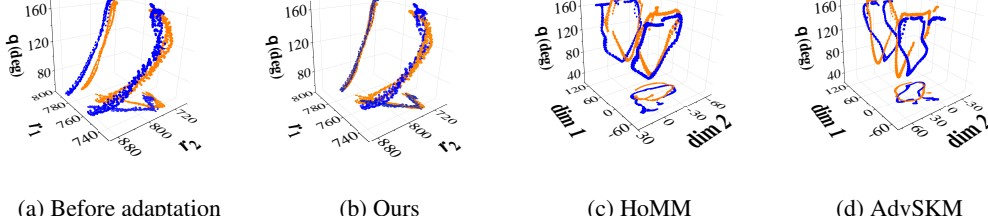

(a) Before adaptation      (b) Ours      (c) HoMM      (d) AdvSKM

Figure 4: Original and adapted data distributions of our method, HoMM and AdvSKM. Note that due to method characteristics, our method is aligned directly using the sensor reading, while the latter two are aligned in the latent space. "dim 1" and "dim 2" are obtained using t-SNE.

leading the network optimization toward the optimal point. As shown in Figure 6, i) before adaptation, the outputs of different regressors significantly differ from each other, leading to a high sequence discrepancy; ii) after adaptation, the discrepancy among regressors reduced and the accuracy of joint angle estimates for all regressors was improved significantly at the same time.

**Fourier Feature Encoding**    Existing literature shows the existence of spectral bias: the tendency of neural networks to prioritize learning low-frequency functions [48]. Figure 7a shows the frequency distribution difference between the prediction and ground truth. Without the component of FFE, the network output shows a large deviation in the region of low-frequency domain from the ground truth (on the left side of this plot). This over-emphasis on the low-frequency components inevitably dominates the prediction capability and weakens the reconstruction of the high-frequency signals.

Our experiment confirms that Fourier Feature Encoding (FFE) reduces the spectral bias of neural networks. Therefore, as Figure 7 shows, our method can approximate the mapping between sensor readings and joint angles in a more faithful way.

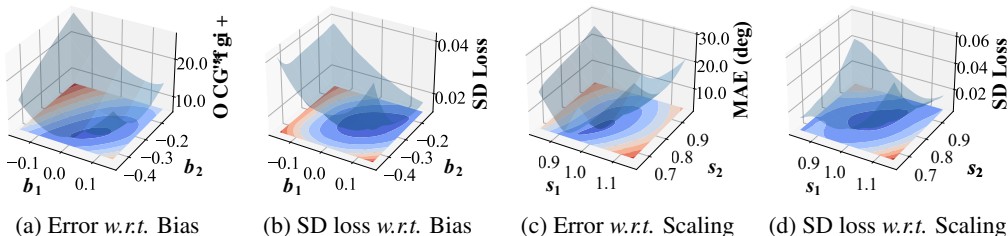

(a) Error *w.r.t.* Bias      (b) SD loss *w.r.t.* Bias      (c) Error *w.r.t.* Scaling      (d) SD loss *w.r.t.* Scaling

Figure 5: Rationale of our Sequence Discrepancy (SD) loss ($c$=5, $l$=0). Our SD loss shares similar minima with the MAE errors. Error: errors of joint angle estimation. As defined in Eq. 2, Bias $B = [b_1, b_2]^T$ and Scaling $W_s = diag\,[s_1, s_2]$.

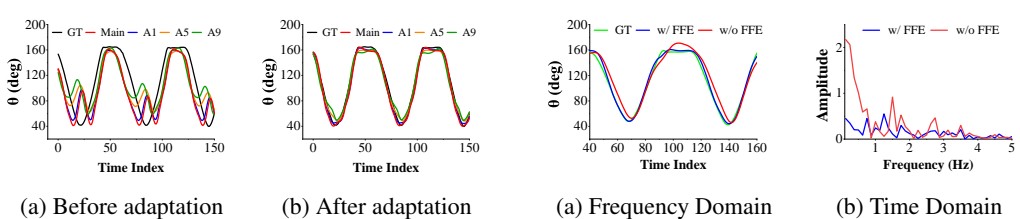

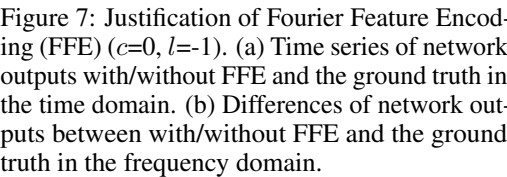

(a) Before adaptation      (b) After adaptation         (a) Frequency Domain      (b) Time Domain

Figure 6: Justification of Sequence Discrepancy Loss ($c$=5, $l$=0). The accuracy of joint angle estimation improves along with the reduction of discrepancy among regressors. A1, A5, A9: auxiliary regressors 1, 5, 9. Main: main regressor. GT: ground truth.

Figure 7: Justification of Fourier Feature Encoding (FFE) ($c$=0, $l$=-1). (a) Time series of network outputs with/without FFE and the ground truth in the time domain. (b) Differences of network outputs between with/without FFE and the ground truth in the frequency domain.

# 6 Limitations and Future Work

**Limitations** The performance of the proposed method is dependent on the quality of the data in both the training and test sets. When the data quality is poor (*e.g.*, noisy), the error in joint angle prediction may still be high even if the sensor readings are well aligned (see Figure 8a and 8b).

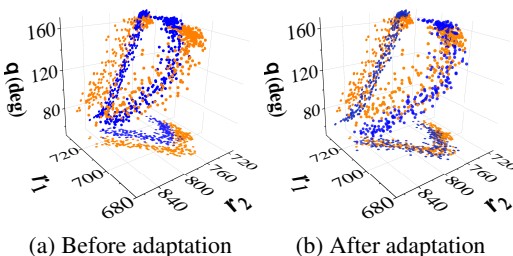

(a) Before adaptation      (b) After adaptation

Figure 8: Failure cases (noisy data). $x$-axis and $y$-axis ($[r_1, r_2]$): sensor readings. $z$-axis: joint angles. Blue and Orange: displacements in the training and test sets, respectively.

Specifically, according to the frequency characteristics of elbow movement, we define the sensor signal greater than 5Hz as noise, and calculated the signal-to-noise ratio (SNR) for the sensor signals within the 11 data sets collected from a single user. Furthermore, we compiled the mean absolute error (MAE) for each set after implementing our adaptive approach. It can be found that our proposed method exhibits significant effectiveness when the SNR exceeds 10 dB. Conversely, maintaining satisfactory outcomes becomes challenging when the SNR falls below 10 dB (see the supplementary materials for the visualization).

**Future Work** As abovementioned, we believe that extending our method to learning with noisy data would be interesting future work. In addition, our work opens up a number of research directions for future efforts. First, in most domain adaptation methods, measuring the similarity between the distribution of the target domain and the source domain is a key step. While in our work, we found that the output of the auxiliary regressors can be used as an indicator of data distribution shifts from the source domain, and verified the feasibility of domain adaptation using only target domain data in flexible sensor applications. In future work, we hope to implement this idea in more diverse applications. Second, although we have only used translation and scaling, this work proves affine-based domain adaptation to be a promising solution to mitigate the data distribution shifts of flexible sensors. The effectiveness of rotation and reflection remains to be studied in our future work.

# 7 Conclusion

Our work proposes a novel self-adaptive motion tracking network to address the challenging data distributional shifts caused by on-body displacements of flexible sensors. To mitigate the effects of such displacements, we propose three novel techniques. First, we propose an Affine Transformation layer that can remap shifted data distributions to those in the training set efficiently. Second, we propose a Fourier-encoded LSTM network that can learn richer frequency components in the input signals and thus improves the accuracy of joint angle estimation. Finally, we propose a Sequence Discrepancy loss equipped with auxiliary regressors that can adapt the parameters of Affine Transformation effectively in an unsupervised manner. Experimental results show that our method can effectively adapt to unknown displacements of flexible sensors worn at different positions.

# 8 Acknowledgements

This work is supported by National Natural Science Foundation of China (62072383), the Fundamental Research Funds for the Central Universities (20720210044), and partially supported by Royal Society (IEC \NSFC \211022).

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
