# Supplementary Materials
# Self-Adaptive Motion Tracking against On-body Displacement of Flexible Sensors

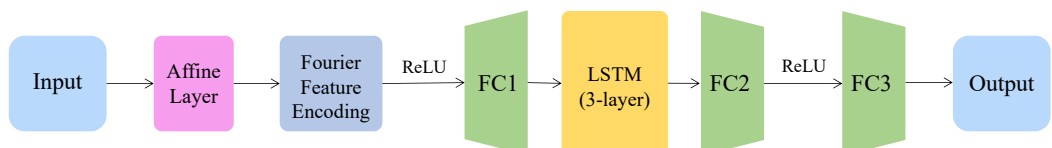

Figure 1: Detailed network architecture. LSTM: Long Short-Term Memory. FC: Fully-Connected.

## 1 Network Architecture

Fig. 1 shows the details of the proposed network's architecture. Specifically, we use a 3-layer LSTM network with a hidden layer size of 256 for sequence feature extraction, and feed both its hidden state and cell state into FC2 for subsequent computation. The output sizes of network layers are listed in Table 1.

| Layer | Output Size |
|---|---|
| Affine Layer | 2 |
| Fourier Feature Encoding | 4 |
| FC1 | 128 |
| LSTM | $3 \times 256 \times 2$ |
| FC2 | 128 |
| FC3 | $10^*$ |

Table 1: Details of network layers. $10^*$: FC3 is an efficient implementation of the $n = 10$ regressors (1 main regressor, $n - 1$ auxiliary regressors) proposed in the main paper. Each of its output dimensions corresponds to one regressor.

## 2 Implementation of SOTA Methods

To make a fair comparison, we used the same architecture as shown in Fig. 1 without the three novel techniques proposed (Affine layer, Fourier Feature Encoding and our Sequence Discrepancy loss equipped with auxiliary regressors). In addition, we used the same loss $\mathcal{L}_{mse}$ (please see the main paper) for the supervised pre-training but made the following modifications for different competitors:

- **MMD** [1]: We applied an Max Mean Discrepancy (MMD) loss at the output of FC2 for the adaptation.
- **D-CORAL** [2]: We applied a CORAL loss at the output of FC2 for the adaptation.
- **DAAN** [3]: We added a softmax layer after FC2 and applied a DAAN loss at its output for the adaptation. The dynamic factor was updated every epoch.
- **DSAN** [4]: We applied a Local Maximum Mean Discrepancy (LMMD) loss at the output of FC2 for the adaptation.
- **HoMM** [5]: We applied a HoMM loss at the output of FC2 for the adaptation.
- **AdvSKM** [6]: We applied a gradient reverse layer, tow multi-kernel layer and a MMD loss at the output of FC2 for the adaptation.

37th Conference on Neural Information Processing Systems (NeurIPS 2023).

# 3 Choice of Hyperparameters

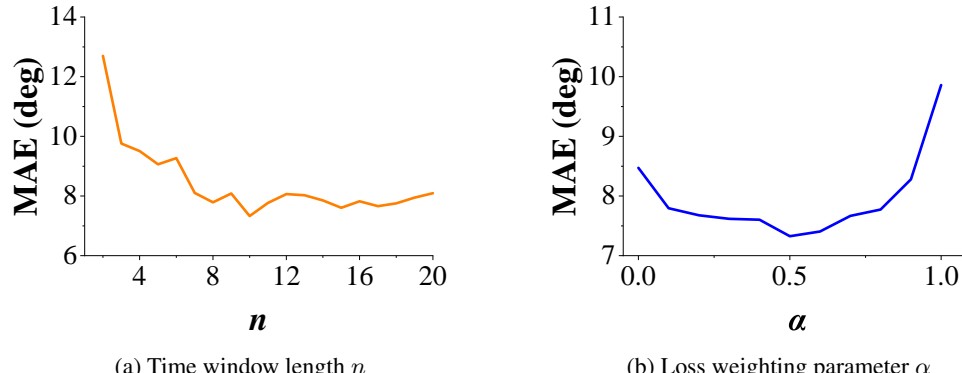

(a) Time window length $n$          (b) Loss weighting parameter $\alpha$

Figure 2: Choice of hyperparameters. (a) Time window length $n$. We set $\alpha = 0.5$ for the evaluation. (b) Loss weighting parameter $\alpha$. $\alpha = 0$ means only $\mathcal{L}_{shape}$ is used; $\alpha = 1$ means only $\mathcal{L}_{mean}$ is used. We set $n = 10$ for the evaluation.

**Time window length** $n$. As Fig. 2a shows, the MAE drops quickly with the increase of $n$ and becomes relatively stable when $n > 8$ and achieves its minimum when $n = 10$. Thus, we choose $n = 10$ in our work. Please note that we use $n - 1$ auxiliary regressors accordingly.

**Loss weighting parameter** $\alpha$. As Fig. 2b shows, the MAE achieves its minimum when $\alpha = 0.5$. Thus, we choose $\alpha = 0.5$ in our work.

# 4 Additional Details of Data Collection

**The 11-displacements dataset.** The dataset contains 5,310 frames (50 fps) of sensor reading and corresponding joint angle data collected from the same participant, covering 11 different on-body displacements. For each displacement, we collected data for 8 consecutive elbow flexion, whose average length is 483 frames (9.66 s). Table 2 shows the detail of these displacements.

Table 2: List of on-body displacements.

| Index | Circular | Lateral |
|-------|----------|---------|
| 1 | -5 | -2 |
| 2 | -5 | 0 |
| 3 | -5 | 1 |
| 4 | -5 | 2 |
| 5 | 0 | -2 |
| 6 | 0 | -1 |
| 7 | 0 | 0 |
| 8 | 0 | 1 |
| 9 | 5 | -2 |
| 10 | 5 | -1 |
| 11 | 5 | 0 |

**Body Profile Details of Five Additional Paticipants** To verify the effectiveness of our method against user diversity, We additionally recruited 5 participants for data collection, including 3 males and 2 females. The body profile details are given in Table 3.

Table 3: Body profile details of the five new participants

| User ID | Gender | Height (cm) | Weight (kg) | Arm circumference (cm) |
|---|---|---|---|---|
| 1 | Male | 179 | 84 | 30.2 |
| 2 | Male | 175 | 86 | 28.5 |
| 3 | Male | 177 | 75 | 27.6 |
| 4 | Female | 172 | 67 | 27.0 |
| 5 | Female | 168 | 65 | 26.6 |

## 5 Influence of sensor signal noise

To gain a deeper understanding of how sensor signal noise affects the performance, we made the following scatter plot. It can be observed that when the SNR of the sensor signals is below 10 dB, the reliability of the proposed method may be challenging to ensure.

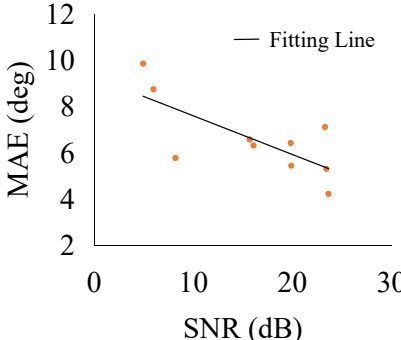

Figure 3: Scatter Plot of SNR vs. MAE. Based on the reasonable range of elbow flexion movement frequencies, we define sensor signals above 5 Hz as noise.