# OpenReview forum: "Self-Adaptive Motion Tracking against On-body Displacement of Flexible Sensors"
_NeurIPS.cc/2023/Conference — NeurIPS 2023 poster_

### Official Review · Reviewer_Bo9S · 2023-07-01

**Soundness:** 3 good
**Presentation:** 4 excellent
**Contribution:** 2 fair
**Rating:** 4
**Confidence:** 3

**Summary:**

In the context of ubiquitous sensing, the paper addresses the issue that on-body devices cannot be firmly worn at a fixed position across different sessions by adapting to unknown displacements. The authors propose three main contributions: (1) A transformation layer adapts to unknown displacements, (2) an LSTM networks identifies patterns, and (3) a discrepancy loss for unsupervised learning.

**Strengths:**

- The problem setup is well motivated. The addressed problem exists in many application areas and research fields.
- The paper is well written and easily understandable.
- The soundness of the technical claims are good and the experiments support the proposed method.
- The authors will make the datasets and source code publicly available.


**Weaknesses:**


- Section 2.3 contains introductory applications that is repeated from Chapter 1, and hence, can be integrated in the Introduction.
- Experiments are only performed on this specific dataset. It is not clear, whether the proposed method can be generalized to other applications, e.g., on datasets such as HHAR, UCI HAR, WISDM, and uWave (time series domain adaptation datasets).
- Experiments in Sec. 5.1 should also cover data from more than one participant
- References are missing and state-of-the-art methods on which experiments are performed are limited. The following methods can also be considered for comparison: MMCD [1], MMDA [2], DDC [3], DAN [4], HoMM [5], CDAN [6], DIRT-T [7], CoDATS [8], AdvSKM [9], and Sinkhorn methods [10,11].

In my opinion, the paper needs an extensive re-write if published at an ML-venue such as NeurIPS. From a methodological perspective the main contribution is the novel sequence discrepancy loss (as (1) learnable affine transformations and (2) Fourier-encoded LSTMs or similar ideas already exist [12]). The paper should be re-written around this main contribution and the on-body placement use-case should be one dataset of many time series data set where the method should be applied on.

**Some minor points:**
- L34ff: But if the features are transformed accurately they still share the same feature space, right?
- L101: it is uncommon to name a specific journal of a citation within the main text
- L115: increase[s]

**Missing references:**
- [1] Naimeh Alipour and Jafar Tahmoresnezhad: Heterogeneous Domain Adaptation with Statistical Distribution Alignment and Progressive Pseudo Label Selection. In Applied Intelligence, volume 52, pp. 8038–8055, October 2021. doi: 10.1007/s10489-021-02756-x.
- [2] Mohammad Mahfujur Rahman et al.: On Minimum Discrepancy Estimation for Deep Domain Adaptation. In Domain Adaptation for Visual Understanding, Springer, Cham., January 2020. doi:10.1007/978-3-030-30671-7_6.
- [3] Eric Tzeng et al.: Deep Domain Confusion: Maximizing for Domain Invariance. In arXiv preprint arXiv:1412.3474v1 [cs.CV], December 2014.
- [4] Mingsheng Long et al.: Learning Transferable Features with Deep Adaptation Networks. In Proc. of the Intl. Conf. on Machine Learning (ICML), volume 37, pp. 97–105, July 2015. doi: 10.5555/3045118.3045130.
- [5] Chao Chen et al.: HoMM: Higher-Order Moment Matching for Unsupervised Domain Adaptation. In Proc. of the AAAI Conf. on Artificial Intelligence (AAAI), volume 34(4), pp. 3422–3429, April 2020. doi: 10.1609/aaai.v34i04.5745.
- [6] Mingsheng Long et al.: Conditional Adversarial Domain Adaptation. In Advances of Neural Information Processing Systems (NIPS), volume 31, pp. 1647–1657, December 2018. doi: 10.5555/3326943.3327094.
- [7] Rui Shu et al.: A DIRT-T Approach to Unsupervised Domain Adaptation. In Intl. Conf. on Learning Representations (ICLR), 2018.
- [8] Garrett Wilson et al.: A Survey of Unsupervised Deep Domain Adaptation. In Proc. of the ACM Trans. on Intelligent Systems and Technology (TIST), volume 11(5), pp. 1–46, October 2020. doi: 10.1145/3400066.
- [9] Qiao Liu and Hui Xe. Adversarial Spectral Kernel Matching for Unsupervised Time Series Domain Adaptation. In Proc. of the Intl. Joint Conf. on Artificial Intelligence (IJCAI), pp. 2744–2750, August 2021. doi: 10.24963/ijcai.2021/378.
- [10] Felix Ott et al.: Domain Adaptation for Time-Series Classification to Mitigate Covariate Shift. In Proceedings of the ACM International Conference on Multimedia (ACMMM), pages 5934–5943, Lisboa, Portugal, October 2022. doi:10.1145/3503161.3548167.
- [11] Huan He et al.: Domain Adaptation for Time Series Under Feature and Label Shift. arXiv preprint arXiv:2302.03133, February 2023.
- [12] Zhang et al.: Learning Long Term Dependencies via Fourier Recurrent Units. ICML 2018.

**Questions:**

none.

**Limitations:**

The limitation of this work is that it is not clear whether the proposed approach can be applied to other time series domain adaptation applications or the method is specifically designed for this on-body displacement of flexible sensors and overfitted on these dataset.

---

> ### Comment · Reviewer_Bo9S · 2023-08-14
> **Re: Rebuttal**
>
> Supporting what reviewer ncj6 mentioned: please address the rebuttal to each review - this makes it easier for the reviewer to enter the discussion based on the own review.
>
> Now to the point: thank you for the rebuttal and the provided PDF. My main criticism about additional participants is resolved (to some degree).
>
> However, I have still an issue with the experiments and specificity of the dataset/data/application (which is brought by reviewer ncj6 also). Imho the experimental results are still a bit weak (#participants, modalities, applications) and to me it is hard to judge the actual impact on the method beyond from what is specifically done here (I share the critic of reviewer fsrB who refers to a "very narrow application of domain adaptation", which "could it be extend to other settings or problems" - but which I cannot judge.
>
> I know it is framed as an "application paper" but I am not sure about the actual impact of this work.

---

> > ### Comment · Reviewer_Bo9S · 2023-08-15
> >
> > Thank you again for the updated experimental results. But my point is/was not the statistical significance of the improvement over the state of the art.
> >
> > My criticism is about a small number of modalities and applications, and a limited number of participants. It is very difficult to judge the impact of the work and how it generalizes to other setups given the provided experimental results.

---

> > ### Author Response · Authors · 2023-08-18
> > **Re：Reviewer Bo9S**
> >
> > *Clarification: We would like to clarify that our previous response was intended to address the issue raised by reviewer ncj6, please see below for our dedicated response to your questions.*
> >
> > Thanks for your constructive feedback and for acknowledging that the principal concern regarding the inclusion of additional participants has been partially addressed. Regarding the remaining concerns, we address them as follows:
> >
> > ### 1) Dataset Richness
> >
> > In flexible sensor studies, participants usually range from one to ten, and data samples range from thousands to hundreds of thousands [1][2][3][4][5], based on task complexity and sensor data dimension. Generally, complex tasks with high-dimensional data require larger datasets (refer to the table below).
> >
> > |   Study  |  Task   | Sensor data dimension | Participants included in dataset|    Dataset size   |
> > |:------:|:------:|:-----:|:-----:|:------:|
> > |  Bian, 2021 [1] | Hand gesture recognition| 4 | 1  | 3.5k |
> > |  Kim, 2018 [2]  | Full-body motion tracking | 10 | 1  | 2.3k |
> > | Zhang, 2022 [3]    |     Gait recognition | 2 | 5  | 0.75k  |
> > | Frediani, 2021 [4] | Body ﬂexion and torsion estimation |  2   |    5   |  Non-public |
> > | Glauser, 2019 [5]  |   Hand pose estimation |  44   | 10 | 105k |
> >
> > In our application, we predict joint angles using data from two sensors, with a dimension of 2 and low task complexity (only one rotational degree of freedom). Hence, our dataset (5 participants, 80,000+ samples) adheres to research norms.
> >
> > ### 2) The actual impact of this work
> >
> > In recent years, flexible sensors have been incorporated into various wearable devices like gloves [5][6], jackets [7], wristbands [1], etc., achieving specific functions through machine learning. Their application has significantly impacted diverse fields including materials science, sensor technology, machine learning, AI, computer graphics, and human-computer interaction. For example, 2019 Nature Technology perspective advocated placing sensors approximately on the body, reducing the need for expert placement [10]. However, collecting datasets covering diverse wear positions for model training increases data costs. Changes in device design also led to additional data collection, consuming time and resources.
> >
> > The contribution of our paper lies in proposing an adaptive method for on-body positioning of flexible sensors. Additionally, introducing the concept of data distribution shifts due to varying sensor positions enhances flexible sensor design optimization, further advancing their application in wearables.
> >
> > ### 3) How it generalizes to other setups given the provided experimental results.
> >
> > We rigorously assessed the generalizability of the proposed method using the DIP-IMU dataset. Although the IMU data present in the DIP-IMU dataset [9] significantly different from flexible strain sensor data (3×3 Unit orthogonal matrix vs. 1-d capacitance value), our method maintains its efficacy (for further details, kindly refer to Table 2 in the attached PDF and our comprehensive response to reviewer ncj6).
> >
> > ### 4) Rebuttal Format
> >
> > We fully endorse the suggestion to "please address the rebuttal to each review." We will address the rebuttal to each review in the upcoming discussions.
> >
> >
> > [1] Bian, Sizhen, and Paul Lukowicz. "Capacitive sensing based on-board hand gesture recognition with TinyML." Adjunct Proceedings of the 2021 ACM International Joint Conference on Pervasive and Ubiquitous Computing and Proceedings of the 2021 ACM International Symposium on Wearable Computers. 2021.
> >
> > [2] Kim, Dooyoung, et al. "Deep full-body motion network for a soft wearable motion sensing suit." IEEE/ASME Transactions on Mechatronics 24.1 (2018): 56-66.
> >
> > [3] Zhang, Quan, et al. "Wearable triboelectric sensors enabled gait analysis and waist motion capture for IoT-based smart healthcare applications." Advanced Science 9.4 (2022): 2103694.
> >
> > [4]Frediani, Gabriele, et al. "Monitoring flexions and torsions of the trunk via gyroscope-calibrated capacitive elastomeric wearable sensors." Sensors 21.20 (2021): 6706.
> >
> > [5] Glauser, Oliver, et al. "Interactive hand pose estimation using a stretch-sensing soft glove." ACM Transactions on Graphics (ToG) 38.4 (2019): 1-15.
> >
> > [6] Gosala, Nikhil Bharadwaj, et al. "Self-Calibrated Multi-Sensor Wearable for Hand Tracking and Modeling." IEEE Transactions on Visualization and Computer Graphics (2021).
> >
> > [7] Zhou, Bo, et al. "MoCaPose: Motion Capturing with Textile-integrated Capacitive Sensors in Loose-fitting Smart Garments." Proceedings of the ACM on Interactive, Mobile, Wearable and Ubiquitous Technologies 7.1 (2023): 1-40.
> >
> > [8] Someya, T., Amagai, M. Toward a new generation of smart skins. Nat Biotechnol 37, 382–388 (2019). https://doi.org/10.1038/s41587-019-0079-1
> >
> > [9] Huang, Yinghao, et al. "Deep inertial poser: Learning to reconstruct human pose from sparse inertial measurements in real time." ACM Transactions on Graphics (TOG) 37.6 (2018): 1-15.

---

### Official Review · Reviewer_fsrB · 2023-07-04

**Soundness:** 3 good
**Presentation:** 2 fair
**Contribution:** 2 fair
**Rating:** 4
**Confidence:** 4

**Summary:**

This paper shows an approach for adaptive learning to track motion trajectories from elbow pad sensor, especially concentrating on modelling of sensor displacements in unsupervised manner during the operation. Tracking method is based on multi-layer neural network architecture with learnable affine layer, Fourier feature encoded LSTM, and multiple outputs regression layers. Furthermore, novel sequence discrepancy loss is introduced to reduce data shift caused by sensor displacements. Proposed techniques are evaluated on collected real datasets with ablation study and comparison to other domain adaptation methods from the literature.

**Strengths:**

This is application oriented paper with novel combination of existing techniques (i.e., affine transformation layer for data shift, Fourier LSTM for pattern matching, and sequence discrepancy loss with aux. regressor for unsupervised/self-adaptive learning) for specific problem, which is evaluated with SOTA comparison and ablation study to work in practice (see also weaknesses). Paper is well-structured and its main processing pipeline is illustrated with related formulation.

Summary of strengths
- Practical solution for (very specific) problem in domain adaptation
- Good combination of different techniques for new application
- Empirical evaluation with comparison and ablation study
- Clearly illustrated processing pipeline and formulation

**Weaknesses:**

Paper only considers very specific problem with one type of sensor and small dataset. It is difficult to assess if this could generalise to different settings. Although there are comparison to SOTA models, paper lacks some basic baseline from signal processing field, e.g., low dimensional data could be estimated using state-space models such as extended kalman or particle filters or at least give more justification of need for learnable methods with adaptive capabilities. Also, most of the SOTA methods presented in the paper are designed for the image data domain adaptation, not for low-dimensional time-series data, so there could be some more related approaches. In addition to utilisation of more comprehensive dataset, paper lack detailed analysis of boundaries in relation to how much noise and sensor displacement it could handle.

Summary of weaknesses
- Very narrow application of domain adaptation (could it be extend to other settings or problems)
- Small datasets (one person, four in supplement)
- Lack of basic baseline (e.g., state space models (EKF, PF etc.))
- Missing the study of data noise and/or boundaries for amount of sensor displacement handled
- Most of the compared SOTA methods designed for different data (e.g., images)

**Questions:**

- How well the prior SOTA methods evaluated in the paper compares to particular time-series type of domain shift?
- What are the boundaries of sensor displacement (i.e., how much noise or data shifts method can handle)?

**Limitations:**

Some of the limitations of proposed approach are shortly analysed in relation to specific issues such as data quality.

---

> ### Comment · Reviewer_fsrB · 2023-08-16
> **Response to rebuttal**
>
> I have read rebuttal and other reviews. I would like to thank authors for additional benchmarking (i.e., with more subjects, additional dataset, and against more related SOTA methods), which definitely improves the original manuscript and results. However, in overall I still find the work quite specific and to show its possibilities, impact, and generalisation capabilities, more work is needed (e.g., experimenting with different modalities, time-series domain adaption problems, and model properties (FFE, Affine Transformation, SD loss) against other  time-series SOTA and baseline models).

---

### Official Review · Reviewer_f2Et · 2023-07-05

**Soundness:** 3 good
**Presentation:** 3 good
**Contribution:** 2 fair
**Rating:** 6
**Confidence:** 3

**Summary:**

Flexible sensors are useful for tracking human status as wearable systems, but they can become displaced when worn, causing challenges for machine learning algorithms. The proposed solution of this paper is a self-adaptive motion tracking network that includes a learnable Affine Transformation layer, a Fourier-encoded LSTM network for pattern identification, and a sequence discrepancy loss with auxiliary regressors for unsupervised tuning of Affine Transformation parameters.

**Strengths:**

1. The proposed method of this paper is useful for real-world applications.
2. The proposed method is intuitive and easy to follow.
3. The experiment results show the effectiveness of their method.

**Weaknesses:**

1. This paper mainly works for the real-world sensor. I would like to know how it works in real-world applications.
2. In the experiments, the author should compare more advanced methods, for example, the least method the author compared is published in 2020. I would like to know how this framework works together with state-of-the-art modules, for example, transformer.

**Questions:**

Please answer 2 of the weakness

**Limitations:**

Please answer 2 of the weakness

---

### Official Review · Reviewer_ncj6 · 2023-07-26

**Soundness:** 4 excellent
**Presentation:** 4 excellent
**Contribution:** 2 fair
**Rating:** 6
**Confidence:** 5

**Summary:**

This paper presents an approach to self-adaptive motion tracking using on-body, flexible sensors that in real world applications may be subject to displacements. The authors present a network that contains a component that automatically learns an affine transformation between training data (with annotations in form of sensor data with their corresponding target joint angles), a Fourier encoded LSTM part for the actual pattern recognition, and a bespoke loss function that uses auxiliary regressors for automated, unsupervised tuning of the parameters of the aforementioned affine transformation. The authors evaluate their method in practical experiments where participants wore elbow sleeves that contain flexible sensors and performed various bend movements. The objective of the evaluation is to measure the accuracy of the predicted joint angles for motion tracking applications. Comparisons to related SOTA models are made and it is shown that the proposed method outperforms all other analyzed models in terms of much better accuracy (distance from ground truth). An ablation study sheds light on the contributions of the individual components of the network and the supplementary material provides details on the implementations, data characteristics etc.


**Strengths:**

This paper is very well written, technically sound, and the experimental results are convincing. I applaud the author for what seems like rigorous work and great thoroughness in execution. Well done. The papers is very application driven, which is fine of course. The tracked problem is relevant for the targeted application domain of motion tracking using body-worn sensors. As such, I can see that the paper may have impact in this field. The authors carefully designed a technical solution in form of a network that specifically, and successfully, addresses domain specific aspects. The experimental evaluation is relevant at least in parts and comparisons to related work / SOTA are done in a very careful and thorough manner. The reported results look promising.


**Weaknesses:**

The experimental evaluation looks a bit limited. While I appreciate the range of displacement that were tackled, the number of participants is a bit small, and so are their individual recordings. I would imagine that the displacements are dependent on a number of factors including the physique of the participants and the tasks performed. Not much is said about either of these and I fear that the promising results may not (or maybe they will — hard to tell!) generalize too much.

On a higher level note: “application papers” typically have a more challenging time at NeurIPS. I do not want to discount the paper merely by the fact that it is mainly an application paper. However, the presented method is very much tailored to motion tracking and I cannot see how the method itself would generalize beyond this. This is not a criticism per se but rather an assessment of the potential impact that paper might have to the broader NeurIPS audience, which would judge as rather limited. I wonder if the authors could branch out their method beyond the specific motion tracking application here? They argue through domain shifts and such, which is a hot topic in the field, but I fail to see how this domain adaptation etc. community would gain much insight from this paper.


**Questions:**

See my comment above re limited impact beyond the particular application domain. I wonder if the authors could elaborate on this a bit.

Also, see my concern about the evaluation. Can the authors provide more information about their participants?


**Limitations:**

Discussion on limitations is there and ok.

---

### Author Rebuttal · Authors · 2023-08-10

# Reviewer #ncj6
## Q1. Limited participants / recording length / physique of participants / task.
To further demonstrate the generalization ability of our method, we conducted additional experiments with five new participants of varying body types (see Table 1 in the uploaded pdf file).
Each participant wore the devices in three on-body positions and performed three distinct physical activities sequentially: __ping-pong__, __basketball__, and __boxing__. In total, we collected 15 (5 participants × 3 on-body displacements) unique data segments across all participants, comprising 81,848 total frames.

Table 2 in the uploaded PDF shows the experimental results of our method on the aforementioned dataset, including two additional SOTA methods, HoMM [1] and AdvSKM [2], as recommended by Reviewers f2Et, fsrB, and Bo9S.
It can be observed that the proposed method demonstrates the lowest average error across data from five users, supporting its applicability to varied user profiles and movement patterns.

## Q2. Branch out proposed method beyond the specific motion tracking application?
To demonstrate the generalizability of our method beyond the task in our paper, we conducted further experiments on the DIP-IMU dataset [3], which includes IMU sensor data from 10 subjects executing diverse motions, along with corresponding 3D human body poses. The objective is to estimate 3D rotations of 15 SMPL model joints using only six IMU readings from head, pelvis, hands, and feet. Similar to DIP [3], we employed an LSTM network for this, yielding 15 joint rotations in axis-angle notation as output.
We trained the model on data from 8 of the 10 subjects, reserving the remaining 2 subjects' data as test set.

As shown in Table 3 in the uploaded PDF, the proposed method achieves slightly improved performance compared to the SOTA HoMM and AdvSKM methods. This demonstrates the strong generalizability of our approach across different input modalities and sensing technologies.

# Reviewer #f2Et
## Q1. How it works in real-world applications.
Our method can be applied in real-world scenarios through the following steps:
1) To begin, users wear the device and perform a flexed-arm calibration pose.
2) Using the collected sensor data, our method fine-tunes the model to adapt to the present on-body device displacement.

## Q2. Comparison to more advanced methods.
Thanks for the suggestion and please see Table 2 in the uploaded PDF for the comparison to two more advanced methods, HoMM [1] and AdvSKM [2], which further demonstrates the superiority of the proposed method.

# Reviewer #fsrB
## Q1. Narrow application of domain adaptation.
Please kindly review our reply to Reviewer #ncj6, Q2, and Table 3 in the provided PDF, showcasing our method's enhanced performance on IMU data. This underscores our approach's robust adaptability across diverse input modes and sensing technologies.
## Q2. Small datasets.
We performed extra experiments on five new participants. Please refer to our response to Reviewer #ncj6, Q1, and Tables 1 and 2.

## Q3. Missing the study of data noise and/or boundaries for amount of sensor displacement handled.
Thank you for your suggestion. We computed SNR for the sensor signals and aggregated MAE for each set of data. As shown in Fig 1 (in the uploaded PDF), our method demonstrates significant efficacy for SNR > 10 dB, while achieving satisfactory outcomes becomes challenging for SNR < 10 dB.
## Q4. Most of the compared SOTA methods designed for different data.
Thank you for your suggestion! To make a fairer comparison, we included comparison results against AdvSKM [2], witch specifically designed for time series data. Please see Table 2 for more details.
# Reviewer \#Bo9S
## Q1. Experiments are only performed on this specific dataset.
Thank you for your comment. Please see our response to Reviewer #ncj6, Q2 and Table 3 in the uploaded PDF, which demonstrates that our method also achieves improved performance on inertial measurement unit (IMU) data.
we carefully assessed the suggested datasets, discovering their focus on classification tasks that our method does not currently accommodate. Hence, we are unable to gauge our approach's performance on these specific datasets.
## Q2. Experiments in Sec. 5.1 should also cover data from more than one participant.
Thank you for your constructive suggestion. We have conducted preliminary experiments with additional participants, and the results indicate that our conclusions remain valid. We will include the updated results from these expanded experiments in the revision.
## Q3. Missing SOTAs.
Thanks for the suggestion and please see Table 2 for the comparison with two suggested SOTAs, HoMM and AdvSKM, which further demonstrates the superiority of the proposed method.
Please note that HoMM and AdvSKM are the two most suitable benchmark methods for our task, which involves unlabeled target domain data and regression modeling.
## Q4. Paper organization.
To clarify, as Reviewer #ncj6 noted, ours is an ''application paper focused on solving an important real-world problem, rather than a ''methodology paper'' aiming to develop a new algorithm applicable to multiple potential applications.
This is within the remit of NeurIPS 2023 Call for Papers,'' Applications (e.g., vision, language, speech and audio)''.

## Q5. Missing references and minor issues.
Thanks for the suggestions. We will include all references in our revision and correct the minor issues.

# References
[1] Chen, Chao, et al. "HoMM: Higher-order moment matching for unsupervised domain adaptation." Proceedings of the AAAI conference on artificial intelligence. Vol. 34. No. 04. 2020.

[2] Liu, Qiao, and Hui Xue. "Adversarial Spectral Kernel Matching for Unsupervised Time Series Domain Adaptation." IJCAI. 2021.

[3] Huang, Yinghao, et al. "Deep inertial poser: Learning to reconstruct human pose from sparse inertial measurements in real time." ACM Transactions on Graphics (TOG) 37.6 (2018): 1-15.

---

> ### Comment · Reviewer_ncj6 · 2023-08-11
> **I have read the rebuttal**
>
> Thanks to the authors for the rebuttal. I appreciate the detailed responses to all questions in general and am satisfied with the responses and additions related to my questions in particular.
>
> As a follow-up: Are those differences between results achieved using SOTA models and yours statistically significant? How did you test (and where are the results)?
>
> Thanks.

---

> > ### Author Response · Authors · 2023-08-15
> >
> > Thank you for your positive feedback and for confirming that our response was satisfactory!
> >
> > Addressing your follow-up question, we repeated the experiments mentioned in our response five times and provided the standard deviations along with the one-way ANOVA test results to justify the statistical significance of our improvement.
> > Please see the tables below for details:
> >
> > Table 1. (a) Mean and standard deviation of motion tracking accuracy on our enlarged dataset comprising data from five additional participants.
> >
> > | User ID |    DAN     |    D-CORAL     |    DSAN    |      HoMM      |   AdvSKM   |      Ours      |
> > |:-------:|:----------:|:--------------:|:----------:|:--------------:|:----------:|:--------------:|
> > |    1    | 17.16±2.18 | **11.84±1.49** | 11.86±1.57 |   11.96±1.73   | 11.84±1.80 |   12.07±1.04   |
> > |    2    | 16.73±1.35 |   16.59±1.81   | 17.04±1.80 | **16.53±1.70** | 16.63±0.95 |   16.69±2.34   |
> > |    3    | 19.60±2.70 |   14.94±5.89   | 13.68±4.19 |   15.24±6.93   | 15.71±6.89 | **12.70±2.79** |
> > |    4    | 23.03±3.11 |   11.76±1.11   | 12.04±1.16 |   12.06±1.49   | 12.08±1.73 | **9.30±2.77**  |
> > |    5    | 19.08±2.72 |   19.41±4.37   | 19.32±4.34 |   20.58±5.88   | 19.44±4.21 | **13.96±2.88** |
> > |   Avg   | 19.12±0.07 |   14.91±0.34   | 14.79±0.22 |   15.27±0.48   | 15.14±0.18 | **12.94±0.25** |
> >
> >
> > (b) One-way ANOVA test on the Avg results (our improvement against SOTA methods).
> >
> > |  Method   |  p-value  | Significance  |
> > |:---------:|:---------:|:-------------:|
> > |    DAN    |  < 0.001  |  Significant  |
> > |  D-CORAL  |  < 0.001  |  Significant  |
> > |   DSAN    |  < 0.001  |  Significant  |
> > |   HoMM    |   0.001   |  Significant  |
> > |  AdvSKM   |  < 0.001  |  Significant  |
> >
> > Table 2. (a) Mean and standard deviation of pose estimation accuracy on the DIP-IMU dataset.
> >
> > | Subject ID |    DAN     |  D-CORAL   |    DSAN    |    HoMM    |   AdvSKM   |      Ours      |
> > |:----------:|:----------:|:----------:|:----------:|:----------:|:----------:|:--------------:|
> > |     9      | 16.27±1.00 | 12.84±0.17 | 13.08±0.16 | 12.98±0.09 | 12.88±0.06 | **12.20±0.12** |
> > |     10     | 16.10±0.59 | 12.08±0.17 | 11.69±0.04 | 11.57±0.13 | 16.63±0.95 | **11.41±0.04** |
> > |    Avg     | 16.18±0.87 | 12.46±0.16 | 12.38±0.10 | 12.28±0.10 | 12.32±0.04 | **11.80±0.06** |
> >
> >
> > (b) One-way ANOVA test on the Avg results (our improvement against SOTA methods).
> >
> > |  Method   | p-value | Significance  |
> > |:---------:|:-------:|:-------------:|
> > |    DAN    |  0.005  |  Significant  |
> > |  D-CORAL  |  0.005  |  Significant  |
> > |   DSAN    | < 0.001 |  Significant  |
> > |   HoMM    |  0.001  |  Significant  |
> > |  AdvSKM   | < 0.001 |  Significant  |
> >
> > These results show that our method makes a statistically significant improvement over SOTA methods with a confidence level of at least 95%.

---

> ### Comment · Reviewer_Bo9S · 2023-08-14
> **See reviewer response below**
>
> I have added my reply to the rebuttal below my review.

---

### Decision · Program_Chairs · 2023-09-21

**Decision:**

Accept (poster)

**Comment:**

Overall this is a borderline paper trending toward accept.  In summary. all reviewers believe that this is a technically sound paper but with limited or specialized interest to the NeurIPS community.  In discussion with the reviewers, no reviewer is entirely against it and because it would bring technical diversity to the conference I would like to see it accepted.